# Cellular Effects of Selected Unsymmetrical Bisacridines on the Multicellular Tumor Spheroids of HCT116 Colon and A549 Lung Cancer Cells in Comparison to Monolayer Cultures

**DOI:** 10.3390/ijms242115780

**Published:** 2023-10-30

**Authors:** Jolanta Kulesza, Ewa Paluszkiewicz, Ewa Augustin

**Affiliations:** Department of Pharmaceutical Technology and Biochemistry, Faculty of Chemistry, Gdańsk University of Technology, 80-233 Gdańsk, Poland; jolanta.kulesza@pg.edu.pl (J.K.); ewa.paluszkiewicz@pg.edu.pl (E.P.)

**Keywords:** unsymmetrical bisacridines, multicellular tumor spheroids, colon and lung cancer cells, cell death, cancer stem cells

## Abstract

Multicellular tumor spheroids are a good tool for testing new anticancer drugs, including those that may target cancer stem cells (CSCs), which are responsible for cancer progression, metastasis, and recurrence. Therefore, we applied this model in our studies of highly active antitumor unsymmetrical bisacridines (UAs). We investigated the cellular response induced by UAs in 2D and 3D cultures of HCT116 colon and A549 lung cancer cells, with an additional focus on their impact on the CSC-like population. We showed that UAs affected the viability of the studied cells, as well as their spherogenic potential in the 2D and 3D cultures. Furthermore, we proved that the most promising UAs (C-2045 and C-2053) induced apoptosis in the HCT116 and A549 spheres to a similar, or even higher, extent than what was found in monolayer conditions. Next, we identified the population of the CSC-like cells in the 2D and 3D cultures of the studied cell lines by determining the levels of CD166, CD133, CD44, and EpCAM markers. We showed that the selected UAs affected the CSC-like population in both of the cell lines, and that A549 was affected more profoundly in 3D than in 2D cultures. Thus, the UAs exhibited high antitumor properties in both the 2D and 3D conditions, which makes them promising candidates for future therapeutic applications.

## 1. Introduction

Despite significant advances in cancer therapy, the mortality rate among cancer patients still remains very high. Recently, a subpopulation of cancer stem cells (CSCs) has been identified, shedding new light on the pathogenesis of cancer. The occurrence of CSCs has been confirmed in a variety of cancer types, including leukemias [1] and solid tumors such as the following: breast [2], pancreas [3], lung [4], prostate [5], and colon [6]. CSCs are a subset of tumor cells with stem cell-like properties, including the ability to self-renew and pluripotency. Therefore, they are responsible for the resistance to conventional chemo- and radiotherapy, as well as act as a key factor in tumor progression, metastasis, and recurrence [7,8,9]. This makes CSCs an attractive target for anticancer therapy in order to not only effectively treat cancer, but also to prevent its relapse. Although there is hope in the development of novel compounds that target CSCs selectively, several drugs that have been shown to be specific for CSCs turned out to be either not very potent or excessively toxic to humans [7,10,11,12]. Some of the current strategies focus on major signaling pathways in CSCs, like the Hedgehog pathway (e.g., vismodegib and sonidegib), the Wnt/β-catenin pathway (e.g., ipafricept and RPI-724), the Notch pathway (e.g., DAPT and MK-0752), and the PPAR pathway (e.g., metformin) [13]. However, these compounds have significant drawbacks. For instance, vismodegib, which has been approved for the treatment of metastatic basal cell carcinoma, only achieves a 48% response rate, and it is associated with various adverse effects in patients [12,14]. On the other hand, metformin, though safer, does not possess the potency needed for effective CSCs targeting, thereby requiring concentrations that exceed physiologically achievable levels [10]. Thus, while some attempts have been made to target CSCs, no effective treatment has yet been established, thereby highlighting the need for new approaches to target these cells through using more potent inhibitors with lower toxicity [7,9]. The discovery of cancer stem cells has revolutionized the understanding of cancer and paved new avenues for research and treatment [15,16]. One of the most recent advances in the study of CSCs is the application of three-dimensional (3D) cultures of cancer cells, which more accurately mimic both the 3D structure of malignant tissue and the microenvironment observed in tumors in vivo. Generally, there are four types of 3D cancer models, two of which are obtained from a single-cell suspension of an immortalized cell line: (a) multicellular spheroids, also known as cancer cell spheroids or multicellular tumor spheroids (MCTS), which are generated in the presence of serum; and (b) tumorspheres (tumor-derived spheroids), which are produced under serum-free conditions. The other two are obtained directly from a tumor tissue: (c) tissue-derived tumorspheres, which are created by the fine slicing of tissue and then partially dissociating it so that it contains, primarily, tumor cells; and (d) organotypic spheroids, which are constructed by cutting tissue into submillimeter pieces and maintaining them in the presence of serum and other supplements but without the dissociation step [17,18,19]. Most studies on CSCs are carried out using tumor-derived spheroids, which are typically enriched in cancer stem cells or cells with stem cell-related characteristics [18]. However, multicellular spheroids—which have a myriad of advantages such as reproducibility, ease of initiation and maintenance, simplicity of genetic manipulation, and clonality of cells—are considered the classical approach for 3D cell cultures. Moreover, MCTS are very well characterized and can simulate the complex conditions present in a patient’s tumor, such as oxygen, metabolic and proliferative gradients, as well as cell–cell interactions. As a result, they show similar therapeutic responses and drug resistance to those observed in vivo [18,19,20].

For many years, our group has been searching for potential anticancer drugs among acridine derivatives. Recently, unsymmetrical bisacridines (UAs) have been synthesized and patented in Europe [21], the USA [22], and Japan [23]. These compounds exhibit high cytotoxic and antitumor activity against many human tumors, mainly those in the lung, colon, prostate, breast, and pancreas [24]. We thoroughly investigated the cellular effects of selected UAs on an HCT116 colon and H460 lung cancer cells, and we showed that these compounds induced apoptosis in the cells of both cell lines; moreover, the induced apoptosis occurred earlier and to a greater extent in H460 compared to HCT116, and this was confirmed by the morphological changes in the nuclei, the presence of a sub-G1 population, active caspase-3, and cleaved PARP along with the annexin V/PI staining and the analysis of the changes in the mitochondrial membrane potential [25]. We have also shown that a noncovalent attachment of UAs to quaternary quantum dots (QDs) improved their cytotoxic activity in lung cancer cells and had protective effects on normal cells [26]. The application of MCTS to our studies on UA derivatives has further revealed the promising properties of these compounds, thereby indicating that UAs are effective against a human colon HCT116 and lung H460 cancer cells that are cultured as spheroids, thus inhibiting their growth and altering spheroid morphology. We also demonstrated that UAs affected the viability of HCT116 cells in both 2D and 3D culture systems [27]. Our experiments on H460 cells showed that spheroids derived from this cell line consisted of almost 50% dead cells, thus making the 3D model of this cell line unsuitable for analyzing the cellular response induced by UAs. Therefore, we looked for another human lung cancer cell line that, like HCT116, would form spheroids with a high percentage of viable cells. We obtained a spherical culture from A549 non-small cell lung cancer cells with a high content of alive cells (almost 90%), so we decided to apply this model to our studies concerning UAs instead of H460 cells. 

Thus, we further investigate here the cellular effects induced by selected UAs (Appendix A) in colon and lung cancer cells in 3D spherical conditions, as well as in 2D monolayer cultures. Two reference compounds were used: irinotecan (IR) and etoposide (ETP). These compounds were also selected for these studies due to their well-established mechanism of action and their common application in the treatment of colon and lung cancer patients, respectively. Additionally, preliminary research was conducted on the impact of unsymmetrical bisacridines on cancer stem cell populations in both 2D and 3D culture models.

## 2. Results

### 2.1. Cytotoxicity of UAs and Etoposide in A549 Cells

The C-2028 and C-2041 derivatives displayed very high levels of cytotoxicity, with their IC_90_ values not exceeding 0.085 µM, while C-2045 and C-2053 were slightly less active with IC_90_ doses of around 0.3 µM (Table 1). The A549 cells were much less sensitive to treatment with the reference compound (etoposide), for which the concentration required to inhibit cell proliferation to 50% (IC_50_ dose) was 5.587 µM. The cytotoxicity of UAs against A549 cells was similar to that previously established in our studies for HCT116, where IC_90_ doses for these compounds were 0.044, 0.049, 0.455, and 0.195 µM, respectively [27]. For the reference compound for the HCT116 cells—irinotecan—the IC_50_ value was established, and it was applied at 4.515 µM.

### 2.2. Morphology and Size of the A549-Derived Spheroids

Since A549 spheroids were applied in our studies for the first time, we had to determine the appropriate seeding density for the generation of the A549 spheres in order to obtain spheroids with a diameter similar to HCT116 (about 400–500 µm). We achieved this by monitoring the initial growth of spheroids from the various densities of A549 cells that were seeded onto a ULA plate (Appendix A). The A549 cells formed spheres with a slightly uneven, jagged periphery, and they were compared to the HCT116 and H460 spheroids [27]. The A549 spheres were substantially smaller when considering the same seeding densities. Therefore, for the formation of the A549 spheroids for further experiments, we selected a seeding density of 2.5 × 10^4^ cells/mL (5000 cells per well).

The microscopic observation and measurement of the diameters of the A549 spheres showed that, during incubation, the growth of the A549 spheroids (Figure 1) was much slower than that in the HCT116 and H460 spheroids [27]. Over time, the A549 spheres increased in size to the least extent, but they did show the greatest ability in retaining their circularity over longer incubation times: almost no visible changes in the periphery of the A549 spheroids were observed, and the spheroids, even on day 14, maintained their compactness and did not evolve into lobular shapes or begin to disintegrate.

The main alteration after treatment with the UAs and etoposide was a considerable inhibition of spheroid growth compared to the untreated control (Figure 1B). The A549 spheres displayed a gradual, significant reduction in their size after only 2 days of incubation with etoposide (ETP), as well as with three of the UAs tested: C-2028, C-2045, and C-2053 (*p* < 0.001). The C-2041 derivative had minimal effect on spheroid size—only a slight inhibition of spheroid growth was observed. In general, the spheroids treated with this derivative exhibited significant growth that was similar to the control (*p* < 0.001). In contrast, treatment with C-2028, C-2045, and C-2053 resulted in smaller spheres compared to those incubated with etoposide. While the reference compound clearly inhibited the spheroids’ growth, it did not cause a reduction in size below the baseline (the size of spheres on day 0—the day of drug administration). However, the spheroids exposed to C-2028, C-2045, and C-2053 after 14 days were 8.53, 18.32, and 18.84% smaller, respectively, compared to their initial size on day 0. 

Except for size, the incubation of A549 spheres with the tested compounds did not result in major changes in the morphology when compared to the control (Figure 1A). Even after 14 days, the circular rim of all the spheroids remained intact, and no visible cells were observed as sprouting at the periphery. Interestingly, it appeared that the A549 spheroids treated with UAs and etoposide for extended periods exhibited slightly more defined edges than the control spheres.

### 2.3. Viability of A549 Cell Cultures in 2D and 3D Conditions after Treatment with UAs and Etoposide 

The A549 cells were cultured in a monolayer, and the spheroids were treated with UAs and etoposide at IC_90_ or 5xIC_90_ doses (IC_50_ or 5xIC_50_ for ETP) for 3 or 7 days, which were then stained with 7-AAD and analyzed by flow cytometry (Figure 2). In order to compare the viability results of the A549 cells in 2D and 3D conditions after UA treatment with the results that we previously reported for HCT116 cells [27], we performed an experiment for a lung cancer cell line under the same conditions that were applied for colon cancer cells. 

In monolayer culture of A549 cells, a 3-day treatment with UAs at IC_90_ doses resulted in a clear increase in the number of 7-AAD^+^ (dead) cells compared to the control, with the fraction of non-viable cells reaching 40.1, 21.0, 23.2, and 28.3% for C-2028, C-2041, C-2045, and C-2053, respectively. For etoposide, this fraction reached only 18.3%. Given the previous experiments on HCT116 [27] and this rather moderate effect on A549 cells, here too we decided to test and compare two variants in 2D and 3D conditions: one with IC_90_ doses (IC_50_ for ETP) with an extended incubation time of 7 days, and the other one with a 3-day treatment time but with a fivefold increase in compound doses to 5xIC_90_ (or 5xIC_50_ for ETP). 

After treatment with UAs for 7 days at IC_90_ doses, the number of non-viable A549 cells was much higher in a 2D culture (about 80–98% of 7-AAD^+^ cells) than in 3D (23–38%), with the exception of the C-2041 derivative (for which this fraction of cells remained similar in both culture conditions (34.9% in 2D and 31.2% in 3D)). When treated for 3 days with C-2028, C-2045, and C-2053 at 5xIC_90_ doses, the proportion of dead cells in the 3D culture (25.9, 39.0, and 36.0%, for C-2028, C-2045, and C-2053, respectively) was very similar to that observed in the 2D culture (22.7, 36.4 and 29.8%), albeit slightly higher. In contrast, for the C-2041 derivative, this fraction was more than 20% lower in the spherical culture compared to the monolayer. In the case of etoposide, whether A549 cells were treated with an IC_50_ dose for 7 days or with a 5xIC_50_ dose for 3 days, the number of 7-AAD^+^ cells was about 10% higher in the 3D than in the 2D culture. Similar to the HCT116 cells [27], we also decided to extend the incubation time with 5xIC_50_ doses from 3 to 7 days in A549 spheroids to see how this would affect the observed cellular response in lung cancer cells. This process was not conducted in the monolayer cultures as it is very challenging to maintain prolonged treatment with high concentrations of compounds. Moreover, even at IC_90_ doses in the A549 adherent model, a very high fraction of 7-AAD^+^ cells was observed after a 7-day treatment. In the spheroids treated with 5xIC_90_ doses of UAs (5xIC_50_ for ETP), extending the exposure time resulted in a marked increase in the number of dead cells—which, for three of the studied UAs (C-2028, C-2045, and C-2053), reached over a 97% rate. Interestingly, a much weaker effect was observed for etoposide, where even after 7-day treatment with a 5xIC_50_ dose, more than 40% of cells remained viable. Similar to previous observations, C-2041 demonstrated the least substantial impact, and treatment with this derivative resulted in the death of less than 37% of cells.

### 2.4. Colony Formation

To determine the potential for HCT116 and A549 cells to return to proliferation, as well as their ability to form colonies following exposure to UAs and irinotecan/etoposide, a colony formation assay was performed. After treatment with the tested compounds, cells were collected, counted, and about 250 cells were incubated for 14 days in a fresh medium, after which their ability to form colonies was evaluated (Figure 3). After being exposed to C-2028, C-2045, and C-2053, the HCT116 and A549 cells showed a complete inhibition of proliferation, which was already noticeable after 24 h of treatment with these compounds. 

Meanwhile, C-2041 only partially blocked cell division, as both HCT116 and A549 cells were able to form colonies even after 120 h of incubation with this derivative, with an average of 6 HCT116 and 16 A549 cells undergoing mitosis. Interestingly, while no HCT116 colonies were observed for any incubation time with irinotecan, the treatment of A549 cells with etoposide did not completely stop the cells’ ability to proliferate, and a few colonies appeared even after 120 h of exposure to the reference compound.

### 2.5. Spherogenic Potential

The ability of HCT116 and A549 cells to generate spheroids following an exposure to the tested compounds was established after the treatment of cells that were cultured in 2D and 3D conditions with IC_90_ doses of UAs and IC_50_ doses of the reference compounds for 3 days. Pictures of the formed spheroids on day 0 (3 days after seeding) and day 14, together with the graphs showing the mean diameters of spheroids obtained, are presented in Figure 4. 

In the case of the HCT116 cells, the control spheroids generated from the cells cultured in 2D and in 3D cultures (secondary spheres) were almost identical in both size and morphology. The A549 cells, however, when cultured as spheroids after being disaggregated into a single-cell suspension were not able to reproduce spherical cultures on day 0. After 14 days of incubation in some cases, spherical forms reappeared, but they were less defined than in the case of spheres that were generated from cells cultured in 2D cultures—the various shapes generated were not perfectly spherical, their periphery was uneven, and slightly lobular shapes appeared. Thus, the correct and definitive measurement of A549 secondary spheres was not entirely possible, particularly on day 0, and the diameters determined using the count and measure extension in cellSens Dimension software version 1.18 were only indicative and approximate. 

After incubation with the tested compounds generally, the spheroids generated were smaller and grew less in time when treatment was applied in the 2D cultures of HCT116 and A549 cells compared to the 3D cultures. HCT116 cells were able to form spheroids that were approximately 250–300 µm in diameter after exposure in 2D cultures to C-2028, C-2045, C-2053, and IR, and, in the case of C-2045, C-2053, and IR, they did not grow in time. The spheroids generated after treatment with C-2041, although much smaller than the control spheres at day 0 (about 330 µm versus 485 µm), grew to similar sizes over time (780 µm for C-2041 and 830 for the control). The HCT116 secondary spheroids after UA treatment for all compounds except C-2041 were considerably smaller than the control on day 0; however, by day 14, spheroids formed after treatment with C-2028, and the IR reached similar diameters to the control. The lowest growth of HCT116 secondary spheroids was observed after exposure to the C-2053 derivative, which was followed by C-2045. 

As with HCT116, in the A549 cells cultured in 2D cultures, treatment with the C-2041 derivative did not considerably inhibit the spherogenic potential of the cells—the spheroids generated were similar in size to the control. The spheroids formed after incubation with C-2045 were slightly smaller than the control on day 0, but over time they became more compact, and their size decreased substantially; meanwhile, the periphery became much more defined. A similar effect was observed for the spheres obtained after exposure of the A549 cells in 2D cultures to etoposide. Interestingly, treatment with C-2028 and C-2053 derivatives completely inhibited the spherogenic potential of A549 cells that were cultured as monolayers, and no spherical forms were observed even on day 14. As mentioned above, secondary sphere formation in the case of A549 cells was difficult to analyze. Nevertheless, some general observations could be made. Firstly, as in the control, after treatment with all of the tested compounds, the A549 cells were unable to recreate spheroids after the primary spherical forms were disintegrated. However, on day 14, some differences were observed between the control and the spheres obtained after treatment with the tested compounds. Most notably, it is worth mentioning that, as in the 2D conditions, after the treatment of 3D spheroids with C-2053, this derivative completely inhibited the ability of A549 cells to form spheroids. None of the other compounds tested presented such activity, including the reference compound. The A549 secondary spheres formed after treatment with the C-2045 derivative were also considerably smaller than with ETP, as was the case for C-2028, but here the difference in size was less defined. The largest spheroids were generated after treatment with the C-2041 derivative.

### 2.6. Annexin V—FITC/PI Double Staining

To investigate the type and intensity of the UA-induced cellular response, as well as to test whether this response in the studied cells depends on the culture method, we performed a flow cytometry analysis of the changes in the asymmetry and integrity of the cell membrane of HCT116 and A549 cells. The cells were grown in monolayers and as spheroids, and—after 72 h—incubation with selected UA derivatives were conducted as follows: C-2045 and C-2053 were subjected to concentrations corresponding to 5xIC_90_ doses. Regarding the reference compounds, the following was applied: irinotecan and etoposide were subjected to 5xIC_50_ doses, double staining with annexin V/PI was performed, and the cells were subjected to a flow cytometry analysis (Figure 5).

The two UA derivatives were selected due to the most promising results from previous experiments (morphology and size analysis, 7-AAD staining, and spherogenic potential assessment). Doses of 5xIC_90_ (5xIC_50_ for the reference compounds) were chosen based on the analysis of the cell viability after treatment (7-AAD staining).

The HCT116 and A549 controls showed similar levels of dead cells in both culture systems—less than 8%. After incubation with the studied compounds, the number of apoptotic cells increased considerably, both in 2D and 3D cultures. In HCT116 cells cultured in 2D, the apoptotic cell number after treatment with C-2045 and C-2053 derivatives amounted to about 20–25%; for C-2053, it also stayed at a similar level (27.20%) in the spherical model. Interestingly, for C-2045, this fraction of cells doubled, from 20.47 in 2D to 40.23% in 3D cultures. In the HCT116 monolayer culture, the most profound induction of cell death was observed after exposure to the reference compound, with 43.53% of cells undergoing apoptosis. In contrast, when applied in the spherical culture of colon cancer, irinotecan induced apoptosis in only 19.27% of cells. Notably, in the A549 cells, the fraction of annexin V positive cells for both tested UA derivatives was higher in 3D (about 40 and 35% for C-2045 and C-2053, respectively) than in 2D cultures (33 and 25%, respectively). On the other hand, etoposide—for which the percentage of cells with translocated phosphatidylserine in 2D cultures was similar to C-2053 (24.28%) in the spherical culture of A549—induced apoptosis in only 11.10% of the treated cells. 

Interestingly, for both cell lines tested, in the 2D monolayer culture, a higher proportion of late-apoptotic cells was observed after incubation with UAs when compared to the early-apoptotic cells. In the 3D cultures, on the other hand, this proportion was quite the opposite—the vast majority of apoptotic cells were in the early phase of apoptosis. In addition, only about 5–10% of the cells showed both phosphatidylserine translocation and cell membrane disruption; meanwhile, in the 2D cultures, this fraction was about 12–18% for the HCT116 and 10–12% for the A549 cells. In turn, after treatment with UAs, the early-apoptotic cells accounted for about 22–34% of the HCT116 and 25–30% of the A549 cells cultured in 3D; meanwhile, in 2D cultures, only 8% of the HCT116 cells and 15–20% of A549 cells were accounted by early-apoptotic cells. Necrosis was induced in a small percentage of cells that were cultured in 3D, and no considerable difference, when compared to the control, was observed for either cell line. In the monolayer culture of HCT116, a slight increase in the number of PI-positive cells was observed after treatment with the tested compounds, but the number of necrotic cells still did not exceed 6%.

### 2.7. Cancer Stem Cell-like Population

In order to test whether the 3D spherical model of cell cultures obtained in our laboratory differed significantly in the number of cells when compared with cancer stem-cell-like features from the monolayer culture, we determined the level of the following surface biomarkers: CD166, EpCAM, CD44, and CD133. All of these were identified in colon and lung cancer stem cells [28,29] in the HCT116 and A549 cells that were cultured in 2D and 3D conditions. Moreover, we tested two previously selected UAs (C-2045 and C-2053) together with irinotecan and etoposide for their ability to affect cancer stem cells in both culture conditions. The results of the flow cytometry analysis are presented in Figure 6. 

In the HCT116 monolayer culture, the levels of all CSCs markers tested were similar, ranging from 87 to 96%. In the colon spheroids, the number of CD166^+^ and CD44^+^ cells was comparable to the 2D cultures, but, for EpCAM and CD133, the number of positive cells was much lower than in the 2D cultures at 80.4 and 56.6%, respectively (Figure 6B). The A549 cells showed high levels of only two of the CSCs markers tested, CD166 and CD44, and the CD166^+^ and CD44^+^ cells accounted for about 92–96% of the A549 cells in both 2D and 3D culture conditions. In the A549 cells that were grown in a monolayer, 5.1% of cells were CD133-positive; meanwhile, in the 3D cultures, this fraction was even lower—less than 1%. The EpCAM^+^ cells were not identified in either the 2D or 3D cultures of A549 cells, and the percentage of cells with this marker was less than 1.5% in both cases (Figure 6C). 

The treatment of HCT116 cells with the tested compounds resulted in a substantial decrease in the number of cells with CSCs markers in both culture systems. In monolayer culture conditions, incubation with UAs reduced the CD166^+^, EpCAM^+^, and CD44^+^ cells by about 15–20%, while the CD133^+^ cells were reduced to as little as 2.4 and 0.8% for C-2045 and C-2053, respectively. Meanwhile, after the exposure of HCT116 cells in monolayer culture conditions to irinotecan, the number of cells with CSC markers decreased to similar levels for all four of the markers tested, and the number of positive cells after treatment was 67.1, 58.9, 63.1, and 50.6% for CD166, EpCAM, CD44, and CD133, respectively. In 3D culture for CD166 and CD44, the decrease in the number of cells with these two markers was less profound than in the 2D culture after treatment with both UAs and IR. In contrast, for EpCAM and CD133, a stronger reduction in the proportion of cells with these markers was observed after UA and IR treatment in the 3D culture of HCT116 than in the 2D culture, and the number of EpCAM-positive cells reached about 55–60% in the spherical culture of the colon cancer cells, while CD133-positive cells accounted for less than 3% of the cells. 

In the A549 cells, a significant reduction in the number of CD166- and CD44-positive cells was observed in both the 2D and 3D cultures following treatment with UAs (*p* < 0.05). Although incubation with etoposide caused a 15–20% decrease in the cell counts with CSCs markers in monolayer culture conditions, in the spheroids, the number of CD166^+^ and CD44^+^ cells was similar to the control. C-2045 and C-2053, on the other hand, showed a stronger effect in 3D than in 2D cultures, and the cell number with both markers after exposure to UAs in 3D cultures was about 60%; meanwhile, in 2D cultures, it was approximately 10% higher (about 70–73%). 

## 3. Discussion

The popularity of 3D spheroid cell culture systems is increasing rapidly as they are becoming essential tools in cancer drug research. They complement conventional 2D monolayer studies prior to animal testing, which can be significantly reduced thanks to this culture method [30]. Spheroids may serve as a tool for the negative selection of drugs that lose their efficacy in a 3D pathophysiological environment, which, conversely, have the potential to identify compounds that exhibit greater activity in 3D than 2D cultures [20]. 

In our recently published work [27], we showed that the application of the multicellular tumor spheroids derived from HCT116 colon and H460 lung cancer cells is a useful tool in examining the efficiency of new antitumor drugs, such as unsymmetrical bisacridines (UAs). Here, we continued our research regarding the mechanism of action of UAs on HCT116 cells. Additionally, due to the fact that the H460 spheroids consisted of almost 50% dead cells, we obtained spheroids from another non-small cell lung cancer cell line—A549—which has a similar morphology to the HCT116 spheres, albeit with a slightly more irregular shape and a jagged periphery. Importantly, MCTS derived from A549 cells consisted of nearly 90% viable cells, so we decided to apply this model to our studies on UAs. A549 cells, like HCT116 [27], proved highly sensitive to UAs treatment, and the concentrations required to inhibit the proliferation of these cells in a 90% concentration (IC_90_ doses) did not exceed 0.35 µM. However, the reference compound—etoposide—presented significantly lower cytotoxicity against A549 cells with an IC_50_ dose of about 5.6 µM. In contrast, the IC_50_ doses determined for UA compounds did not exceed 0.06 µM). The determined IC_50_ dose for etoposide was comparable to that established in other laboratories [31,32], and it was also close to the IC_50_ for irinotecan in HCT116 cells (about 4.5 µM), which we previously determined [32]. The colony formation assay showed that three of the tested UAs—C-2028, C-2045, and C-2053—already completely blocked the proliferation of the HCT116 and A549 cells after 24 h; meanwhile, after exposure to C-2041, some of the cells underwent mitosis even after 120 h of treatment. 

The spherical cultures provided an appropriate model for testing new antitumor therapeutics, especially those targeting CSCs [8]. In our study, we aimed to compare the levels of CSC markers in the adherent and spherical cultures of HCT116 and A549 cells, as well as aimed to assess the impact of the studied compounds on this population of cells. The selection of adequate CSC markers was a critical and challenging step in determining the proportion of the CSC-like cells in a studied population since none of the known markers were universal and entirely reliable [33]. Therefore, in addition to the frequently used CD133 marker (which has proven to be rather controversial [34,35,36,37]), we chose three more proteins—CD44, CD166, and EpCAM—for CSC identification in our studies. These markers have been reported to be present in lung and colon cancer cells [28,29]. In our study, the levels of all of the selected markers observed in both HCT116 cell culture systems were relatively high and similar to those reported by other researchers [8,29,38,39]. On the other hand, in A549, neither the monolayer nor the spherical model presented a significantly high fraction of EpCAM^+^ and CD133^+^ cells. The literature presents widely divergent data regarding EpCAM expression in A549 cells. For instance, Baharuddin et al. [40] reported that A549 EpCAM^+^ cells account for about 5% of all cells in monolayer culture conditions, while Karimi-Busheri et al. [41] reported 34%. In contrast, Breuninger et al. [42] showed that 96% of A549 cells are EpCAM-positive, which is similar to Liao et al. [43] who stated that almost 100% of cells possess this marker. On the other hand, although CD133 has been listed as a potential marker for CSCs in lung cancer cells, its reported level in A549 cells in most cases was very low, i.e., it did not exceed 2% [36,44,45,46,47] (which was comparable with our results). Thus, determining the levels of several different markers seems to be a useful initial step for any CSC-related investigations. While A549 cells are in monolayer and spherical cultures, the difference in the fraction of cells with CD166 and CD44 markers in our case was not profound (Figure 6C). Moreover, in HCT116, these two markers were also quite similar in both culture conditions. For EpCAM and CD133, there was a much more pronounced discrepancy (Figure 6B). Therefore, it is impossible to state unequivocally whether the 3D model we applied in our research differs significantly in CSC content from monolayer conditions as this clearly depends on the specific markers that are determined. Nonetheless, when analyzing two of the markers that were tested, CD166 and CD44, we found that, in HCT116, the fraction of cells with these markers after treatment with UAs and irinotecan was lower in the 2D than in 3D cultures, thereby indicating that the compounds lose some of their effect on CSCs when applied in spherical cultures. However, it is worth highlighting that, in A549, a more pronounced effect in decreasing the number of CD166^+^ and CD44^+^ cells was observed in spheroids, where the fraction of cells with these markers was about 10% lower than in the adherent model after treatment with C-2045 and C-2053. Importantly, in the case of etoposide, which already has a weaker effect on CSC-like cells in monolayer cultures compared to our selected UAs, its further potency was reduced when applied in spheroids. Thus, C-2045 and C-2053 exhibited promising effects on lung cancer cells, especially given their enhanced efficacy in 3D cultures. 

Most of the research performed on spherical cultures shows that spheres are a more resistant platform to therapeutic agents, and that many compounds have markedly reduced efficacy in a 3D environment compared to a 2D environment [48,49,50]. The differences in drug response between 2D and 3D cultures may be caused by variations in drug penetration, drug gradients, altered gene expression, increased survival signaling, DNA repair, the pH, and transporters associated with drug resistance, as well as maybe also resulting from the mechanism of the action of the drug itself. However, there are reports that some drugs, which are not necessarily very potent in 2D cultures, show their effects only in 3D cultures, as seen in some cases where the molecular target is expressed only or especially in a three-dimensional environment [20]. The determination of cell viability after treatment with UAs once again highlighted the very promising potential of C-2045 and C-2053. When applied at 5xIC_90_ doses, these compounds proved slightly more effective in inducing cell death in A549 cells that were cultured in a 3D environment compared to a 2D environment. In HCT116 cells, a similar but more profound effect was observed for the C-2045 derivative, while, in the case of C-2053, the number of dead cells was slightly lower in 3D than in 2D cultures [27]. The annexin V/PI double staining performed with these two UAs confirmed the results from the cell viability assessment, thus showing that these compounds retained their activity in a 3D environment, or even proved more potent in this culture system than in a monolayer culture. Most notably, both reference compounds induced apoptosis to a much smaller extent in 3D than 2D cultures, with IR losing more than half of its proapoptotic activity when observed in monolayer conditions when applied with spheroids. It is worth mentioning that, in general, in 2D cultures, we observed a predominance of late-apoptotic cells; meanwhile, in spheroids, most of the cells were still in the early stages of apoptosis, which suggests that, due to the compact and dense nature of the spherical cultures along with their large volume, the cell exposure to the compounds was limited and that drug penetration into the deep layers of the cells was hindered. Thus, the effect induced by the studied compounds may be delayed compared to the monolayer, where all cells were evenly exposed to an equal concentration of the drug simultaneously. Nevertheless, the fact that C-2045 and C-2053 remained potent in cell death induction in spherical cultures highlights their promising therapeutic efficacy. This was evident even in the initial evaluation of the UA influence on spherical cultures through the measurement of spheroid diameters after exposure to the compound with increasing times of incubation, where, for both cell lines, the C-2045 and C-2053 derivatives proved to be the most potent UA derivatives and caused a significant reduction in spheroid diameters. 

The ability to generate spheroids after treatment with the studied compounds was established additionally as an extension of the colony formation assay to provide us with further information on the cells’ capacity for self-renewal. Interestingly, although both cell lines were capable of forming spheroids, the secondary spheres after dissociation of primary A549 spheroids did not regenerate after 3 days, thereby suggesting that, during disaggregation into a single-cell suspension, some changes occurred in the A549 cells that affected their spherogenic potential. The ability of cells to form spheroids is influenced by several factors, such as cell type, culture conditions, the presence of growth factors, and other signaling molecules. Since the cell type and culture conditions did not change compared to the spheres obtained from 2D cultures, it seems that some alterations happened in the growth factor or adhesion protein levels, which made it impossible for A549 cells to form secondary spheres. Analysis of the spherogenic potential of HCT116 and A549 cells after treatment with UAs in 2D and 3D conditions showed that, generally, the spherical cultures were less affected by tested compounds than their monolayer correspondents, as a greater capacity to recreate spheres was observed in secondary sphere formation compared to spheres obtained after exposure to UAs in the monolayer. However, C-2053 was the most potent derivative in both culture conditions, and the A549 cells did not form spheres after incubation with this derivative in 2D and 3D cultures; meanwhile, the HCT116 spheres generated after exposure to this compound were the smallest and did not grow in time (2D culture) or grew to the least extent (3D culture). In contrast, the secondary spheres regenerated after treatment with irinotecan and etoposide grew in time, and, in the case of HCT116 spheres after 14 days, they reached similar sizes to the control (780 µm vs. 834 µm for the control). The least effect on the spherogenicity of both cell lines was observed after treatment with the C-2041 derivative, both in the case of primary and secondary spheres. Its distinct structure—a different aminoalkyl linker between the two monoacridine units of which UAs consist [24]—may affect the ability of C-2041 to enter cells, as well as their capacity to penetrate the deeper layers of the spheroids. 

It has been observed lately that the localization of UAs in cells is pH-dependent, and an increased concentration of these compounds in organelles with low pH has also been detected [25,51]. This—along with the fact that pH strongly influences UA protonation states, self-association ratio, and solubility [52]—may provide some insight into why some of these compounds exhibit even more promising results in 3D spheroids, which are characterized by a specific pH gradient that is similar to that observed in vivo. At a physiological pH (between 6 and 8), there are two or three individual protonation forms of UAs, each of which can likely affect the cellular processes differently at the molecular level [52]. Thus, the specific form (or forms) of UA compounds that are present in spheroids may differ from that in monolayer cultures, where the pH does not vary.

## 4. Materials and Methods

### 4.1. Tested Compounds

The unsymmetrical bisacridines (UAs) C-2028, C-2041, C-2045, and C-2053 were synthesized, according to a previously published procedure [24], as methanosulphonians (C-2028, C-2041, and C-2045) or as a monochloride (C-2053) in the Department of Pharmaceutical Technology and Biochemistry, Gdańsk University of Technology. Both stock and working solutions were prepared in sterile deionized Mili-Q water (Merck/Sigma-Aldrich, Darmstadt, Germany). The reference compounds irinotecan (IR) and etoposide (ETP) were purchased from Sigma-Aldrich (St. Louis, MO, USA), and the stock solution was prepared in dimethylsulfoxide (DMSO; POCH S.A, Gliwice, Poland). Meanwhile, the working solutions were prepared in sterile, deionized Mili-Q water (where they had a final DMSO concentration in the culture medium of below 0.3%). 

### 4.2. Cell Lines and Culture Conditions

Human colon carcinoma HCT116 and non-small-cell lung carcinoma A549 cells were purchased from the American Type Culture Collection (ATCC, Manassas, VA, USA). They tested negative for mycoplasma when using the Universal Mycoplasma Detection Kit (ATCC, Manassas, VA, USA). The HCT116 cells were cultured in McCoy’s 5A medium (Sigma-Aldrich, St. Louis, MO, USA), and the A549 cells were cultured in an F-12 Ham Kaighn’s Modification medium (Sigma-Aldrich, St. Louis, MO, USA). Both media were supplemented with 10% heat-inactivated fetal bovine serum (FBS; Biowest, Nuaille, France), 100 µg/mL of streptomycin, and 100 U/mL of penicillin (Sigma-Aldrich, St. Louis, MO, USA). The cells were incubated at 37 °C in a 5% CO_2_ atmosphere. All experiments were performed with the cells in the exponential phase of growth. Both the HCT116 and A549 cells used in the experiments were within the passage range of 3 to 12.

### 4.3. Cell Growth Inhibition Assay

To estimate the cell viability, the 3-(4,5-dimethylthiazol-2-yl)-2,5-diphenyltetrazolium bromide (MTT) assay was used. The HCT116 and A549 cells were seeded in 24-well plates at 2 × 10^4^ cells per well. After 24 h of incubation at 37 °C in a 5% CO_2_ atmosphere, unsymmetrical bisacridines or reference compounds (IR/ETP) were added at concentrations of up to 10 µM for UAs and up to 200 µM for the reference compounds. After 72 h of incubation, the 200 µL/well of MTT (Abcam, Cambridge, UK) at a concentration of 4 mg/mL was added, which was then incubated for 3 h at 37 °C. Next, the culture medium from each well was removed, the formazan crystals were dissolved in DMSO, and an absorbance at 540 nm was measured using a microplate reader (iMark^TM^, Bio-Rad, Hercules, CA, USA). The concentrations of the drugs required for the inhibition of cell growth by 90% (IC_90_) for UAs and 50% (IC_50_) for the reference compounds compared with the untreated control cells were calculated from the curves that plotted the survival as a function of the dose. 

### 4.4. Generation of Multicellular Spheroids

For the spheroid formation, 96-well Corning^®^ Costar^®^ Ultra-Low Attachment (ULA) round-bottomed plates (Corning Incorporated, Kennebunk, ME, USA) were used. The HCT116 spheroids were obtained according to the procedure described previously [27]. Briefly, the cells were trypsinized, counted, centrifuged to remove trypsin, and resuspended in a fresh culture medium. Then, a 200 µL/well of HCT116 cell suspension at a density of 1 × 10^4^ cells/mL (2000 cells per well) was dispensed into the ULA plate, and the plate was centrifuged for 15 min at 1200 rpm at room temperature to initiate cell aggregation. The plate was then incubated at 37 °C in a 5% CO_2_ atmosphere for 3 days to enable the formation of spheroids before further experiments.

To determine the optimal seeding density for A549 spheroid formation cell suspensions prepared adequately according to the method described above, the following densities were applied: 1 × 10^4^, 1.5 × 10^4^, 2 × 10^4^, 2.5 × 10^4^, 3 × 10^4^, and 3.5 × 10^4^ cells/mL (2000, 3000, 4000, 5000, 6000, and 7000 cells/well, respectively). These cell suspensions were then dispensed into an ULA plate. After seeding, the plate was centrifuged (RT, 15 min, 1200 rpm) and then incubated at 37 °C in a 5% CO_2_ atmosphere for 3 days. Afterward, the images of the generated spheroids were taken using a 4x objective in an OLYMPUS IX 83 inverted microscope with an XC 50 camera and cellSens Dimension software version 1.18. Then, 100 µL of the medium in each well was carefully replaced with a fresh medium, and this day was then referred to henceforth as day 0. The images of the spheroids were taken daily for the next three days, the diameter of each spheroid was measured, and the mean value was calculated. For further experiments, the seeding density of 2.5 × 10^4^ cells/mL (5000 cells/well) was chosen, and the spheroids were obtained according to the method described above for HCT116.

### 4.5. A549 Spheroid Size and Morphology Assessment 

The lung cancer spheroids were generated as described above. The analysis of the effect of the UAs and etoposide on A549 spheroids was performed according to the methodology previously described for the same experiment on HCT116 spheres [27]. Briefly, 72 h after seeding, images of each spheroid were taken and a 100 µL of culture medium was replaced with a fresh medium in the control (or a fresh medium with 0.05, 0.085, 0.3, 0.3, and 5.6 µM of C-2028, C-2041, C-2045, C-2053, and etoposide, respectively). Images of at least 8 spheroids for each compound were taken every 2–3 days (up to 14 days) after drug treatment, and the spheroid diameters were measured each time using the cellSens Dimension software. Results were obtained from four independent experiments (*n* = 4), and the mean value of the spheroid growth was calculated as presented below: $$\%\;spheroid\;growth=\frac{d_{x}}{d_{0}}*100\%,$$
where *d_x_* is the mean diameter of at least 8 spheres at a given day of incubation, and *d*_0_ is the mean diameter of at least 8 spheres at day 0 (day of the drug treatment).

### 4.6. Cell Death Assay

The cell death evaluation of A549 cells was conducted similarly to the methodology described earlier for HCT116 cells [27] with 7-aminoactinomycin D (7-AAD) dye (Thermo Scientific, Waltham, MA, USA). Briefly, in the monolayer cultures, 1.2 × 10^6^ of A549 cells were seeded on a 100 mm plate (4 × 10^5^ cells for 3 days and 1 × 10^5^ for 7 days of untreated control), which were then allowed to adhere overnight. The cells were then treated for 3 or 7 days with UAs at concentrations corresponding to IC_90_ values (0.05, 0.085, 0.3, and 0.3 µM for C-2028, C-2041, C-2045, and C-2053, respectively) or 5xIC_90_ values; meanwhile, the etoposide was added at a concentration corresponding to IC_50_ (5.6 µM) or 5xIC_50_. After drug treatment, 0.5 × 10^6^ cells were collected from the plates, centrifuged at 1000 rpm for 5 min at RT, and washed twice with PBS. They were then pelleted, resuspended in 150 µL of PBS, and stained with 7-AAD dye (1 µg/mL) for 15 min in the dark at RT. The A549 spheroids were generated as described above, and then, 72 h after seeding, 100 µL of the culture medium was changed and the cells were treated for 3 or 7 days with drugs at concentrations corresponding to IC_90_ or 5xIC_90_ values for UAs and IC_50_ or 5xIC_50_ values for the reference compounds. After drug treatment, the spheroids were disaggregated to obtain a single-cell suspension for a flow cytometry analysis. To accomplish this, the spheroids were collected, centrifuged, washed with PBS, treated with 200 µL of trypsin, and pipetted to promote cellular detachment. Next, a fresh medium was added to neutralize the trypsin. The cells were then centrifuged, washed twice with PBS, suspended in 150 µL of PBS, and stained with 1 µg/mL 7-AAD for 15 min in the dark at RT. After staining, the cells were analyzed using flow cytometry with FACS Accuri^TM^ C6 (BD, San Jose, CA, USA), and the data were analyzed using BD Accuri^TM^ C6 Software Version 1.0.264.21. 

### 4.7. Colony Formation Assay

The ability of HCT116 and A549 cells to return to proliferation and form colonies after treatment with UAs and the reference compounds was measured by colony formation assay. The cells were exposed to the tested compounds at concentrations corresponding to IC_90_ values (or IC_50_ for IR/ETP), and—after incubation for 24, 72, and 120 h—the cells were harvested and counted, and about 250 cells were seeded in a 6-well plate with a fresh medium, which were then incubated for two weeks. After this time, the colonies were fixed with 80% EtOH and stained with Giemsa dye. Then, the plates were photographed, and the number of the colonies was counted. 

### 4.8. Establishment of the Spherogenic Potential 

To analyze the spherogenic potential, the HCT116 and A549 cells were cultured in 2D (monolayer) and 3D (spheroid) conditions as described above. Then, after treatment with UAs and the reference compounds for 3 days at IC_90_ doses (IC_50_ for IR/ETP), the cells were collected into Falcon tubes, centrifuged, and resuspended in a fresh culture medium. The cells cultured as spheroids were disaggregated into a single cell suspension, as explained in Section 4.6. Then, the cells were counted and 200 µL of an HCT116 cell suspension at 1 × 10^4^ cells/mL or A549 at 2.5 × 10^4^ cells/mL were dispensed into a ULA plate, which was then incubated at 37 °C in a 5% CO_2_ atmosphere for 3 days. After this time, 100 µL of a culture medium was replaced with a fresh medium, and then the plates were incubated for two weeks. Pictures of the spheroids were taken on days 0, 7, and 14, and the diameters of the formed spheres were measured using cellSens Dimension software. 

### 4.9. Annexin V/PI Dual Staining

Changes in the cytoplasmic membrane of the HCT116 and A549 cells were analyzed by flow cytometry using the FITC Annexin V Apoptosis Detection Kit (BD Biosciences, San Jose, CA, USA) according to the manufacturer’s procedure. Briefly, after incubation of the monolayer and spherical cultures of the HCT116 and A549 cells with C-2045, C-2053, and IR/ETP at 5xIC_90_ doses (5xIC_50_ for reference compounds), 1 × 10^6^ cells were centrifuged, washed twice with PBS, pelleted, and then resuspended in 100 µL of a binding buffer that contained Annexin V-FITC and PI (the spheroids were dissociated into a single-cell suspension beforehand). The cells were then incubated for 15 min in the dark at RT, diluted with 400 µL of binding buffer, and analyzed by flow cytometry with FACS Accuri^TM^ C6. The cells stained with only annexin V were considered early-apoptotic, the cells with both annexin V and PI-positive as late-apoptotic, and cells stained only with PI as necrotic.

### 4.10. Identification of Cells with CSC-like Features

The levels of the following surface biomarkers were determined in the 2D and 3D cultures of the HCT116 and A549 cells using flow cytometry analysis: CD44, CD133, CD166, and EpCAM. The cells were cultured in both conditions after treatment with C-2045, C-2053, and IR/ETP at 5xIC_90_ doses (5xIC_50_ for IR/ETP). These were then collected and centrifuged (the spheroids were previously dissociated into single-cell suspensions), washed twice with Stain Buffer BSA (BD Biosciences, San Jose, CA, USA), and then resuspended in a Stain Buffer BSA that contained the following appropriate antibodies: CD44-BB515, CD133-PE, CD166-PE, or EpCAM-BB515 (BD Biosciences, San Jose, CA, USA). Following the incubation for 30 min at 4 °C in the dark, the cells were centrifuged, washed twice with Stain Buffer BSA, resuspended in it, and then subjected to flow cytometry analysis. As a negative control, an isotype-matched labeled control was used for each antibody. 

### 4.11. Statistical Analysis

The results are presented as the means ± SDs from at least three independent experiments. Due to this, all of the statistical analyses were exclusively performed using nonparametric statistics. Differences between the UAs (C-2028, C-2041, C-2045, C-2053) and the reference compounds (irinotecan and etoposide) in comparison to the control group for each studied parameter were analyzed by Dunn’s nonparametric multiple comparisons test, and these were applied only if the nonparametric ANOVA (Kruskal–Wallis test) showed that the variability between the group medians was significantly greater than what was expected by chance. Additionally—to assess the influence of time upon the size of spheroids for the control, UAs, and reference compounds—the Jonckheere test for the ordered alternatives was performed. Finally, the Mann–Whitney U test was used to analyze the differences between the results of the 7-AAD staining experiment for the 2D and 3D samples. Two statistical packages were used as follows: GraphPad InStat v. 3.0 and KyPlot v. 2.0. For all the statistical analyses, the differences of *p* < 0.05 between the two groups were considered statistically significant as per the following: * *p* < 0.05, ** *p* < 0.01, and *** *p* < 0.001.

## 5. Conclusions

To summarize, unsymmetrical bisacridines have shown potential for anticancer therapy due to their efficacy in both 2D and 3D environments. These compounds inhibit HCT116 and A549 spheroid growth, as well as influence the viability of cells in both culture conditions. We demonstrated that incubation with UAs decreased the spherogenic potential of HCT116 and A549 cells, and—in the case of the C-2045 and C-2053 derivatives—the tested cell lines were either incapable of generating spheroids after treatment with these compounds in both 2D and 3D conditions or the spheres formed did not grow over time. Moreover, these promising compounds were also found to induce apoptosis in HCT116 and A549 spheres, with a similar or even higher proportion of cells being affected than in the adherent model. Most importantly, C-2045 and C-2053 significantly affected the cancer-stem-cell-like population in both cell lines, with the effect being even more pronounced in A549 spheroids than in monolayer cultures. However, the experiments concerning CSCs were performed with UAs for the first time, so the results obtained are rather preliminary and require further investigation. Nonetheless, this study sheds greater light on the cellular response induced by UAs in HCT116 and A549 cells that are cultured in 2D and 3D conditions with additional relation to CSCs. Despite the current lack of comprehensive understanding regarding the exact mechanism of action of UAs, our research significantly contributes to expanding the knowledge about the effects of these derivatives on studied cells. This provides a foundation for future applications of these promising compounds in anticancer therapy, thereby offering the potential for the development of improved and more effective treatment regimens. Additionally, our research highlights the value of multicellular tumor spheroids as useful additions in the study of new chemotherapeutic agents, as they can even provide preliminary data on the impact of tested compounds on cancer stem cell population.

## Figures and Tables

**Figure 1 ijms-24-15780-f001:**
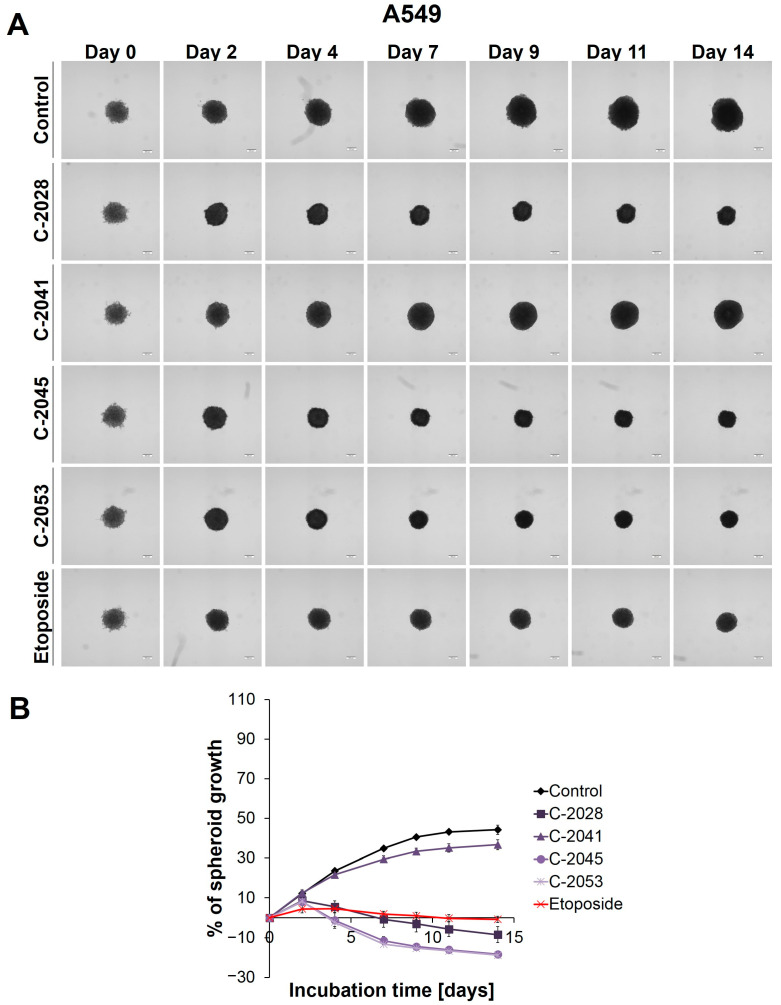
A549 spheroid morphology and kinetics. The spheroids were incubated with UAs at concentrations that corresponded to the IC_90_ values and the etoposide at an IC_50_ dose for 14 days. Every 2–3 days, images of the spheroids were taken, and diameters were measured. (**A**) Representative images of the A549 control spheroids and spheroids treated with tested compounds. (**B**) Growth kinetics shown in a graph form as a percentage of the A549 spheroid growth over time. The data represent the averages of the four independent experiments with standard deviations. Scale bar is 200 µm. (*n* = 4).

**Figure 2 ijms-24-15780-f002:**
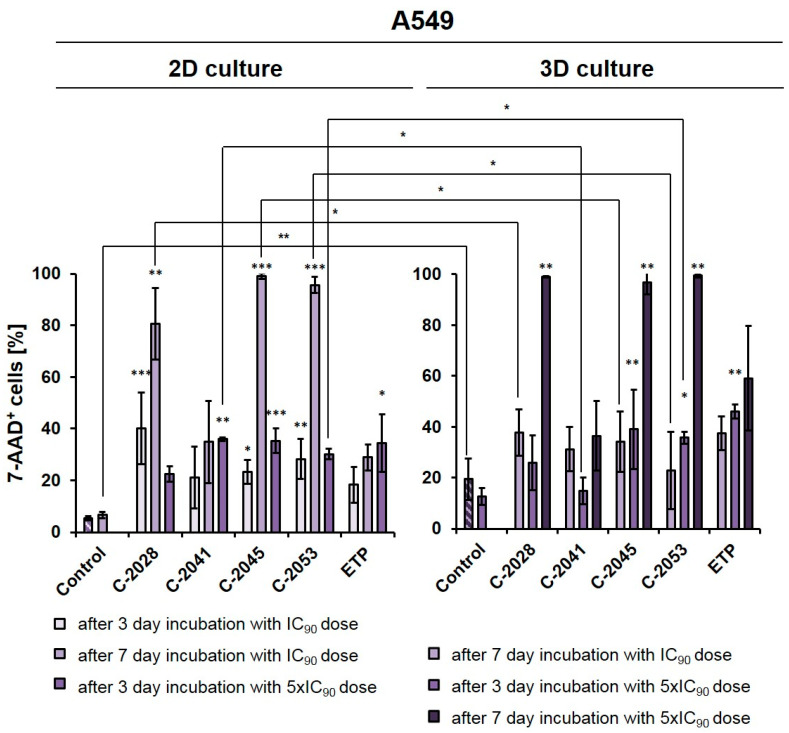
Effects of C-2028, C-2041, C-2045, C-2053, and etoposide (ETP) on cell viability in A549 cells cultured in 2D and 3D conditions. Cells were incubated with tested compounds at concentrations corresponding to IC_90_ and 5xIC_90_ values (IC_50_ and 5xIC_50_ for ETP) for 3 or 7 days, which were then stained with 7-AAD and subjected to flow cytometry analysis. The bar graphs show the quantified data, which are expressed as the percentages of 7-AAD^+^ (dead) cells after incubation with the tested compounds in 2D (**left**) and 3D (**right**) cell cultures. The data are presented as the means ± SDs of three to eight independent experiments. Statistical analysis was performed using Dunn’s nonparametric multiple comparisons test and the Mann–Whitney U test. Significant difference was acknowledged at * *p* < 0.05, ** *p* < 0.01, and *** *p* < 0.001. (*n* = 3–8).

**Figure 3 ijms-24-15780-f003:**
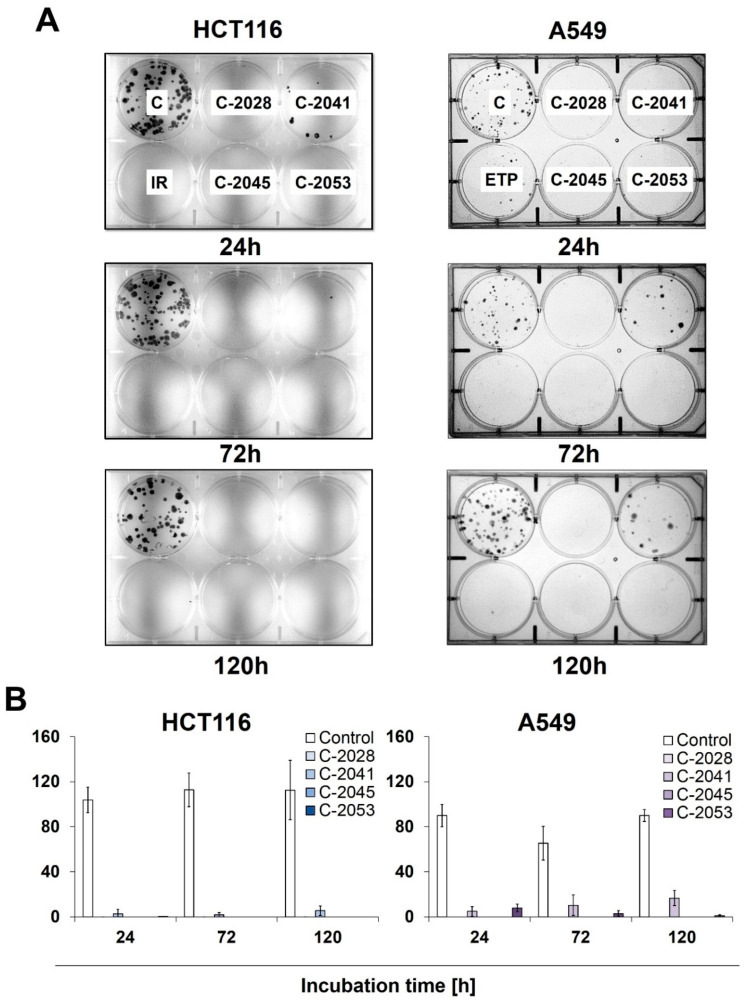
The ability of HCT116 and A549 cells to return to proliferation after UA exposure. Cells were treated with IC_90_ doses of C-2028, C-2041, C-2045, and C-2053 compounds (IC_50_ for irinotecan and etoposide). After the indicated drug exposure, approximately 250 cells were cultured for two weeks in a fresh medium, and the number of colonies was counted. (**A**) Representative pictures of HCT116 (**left**) and A549 (**right**) cells after post-incubation and Giemsa staining. (**B**) Bar graphs show the quantified data, and these are expressed as the number of colonies ± SDs for HCT116 (**left**) and A549 (**right**) cells. (*n* = 3).

**Figure 4 ijms-24-15780-f004:**
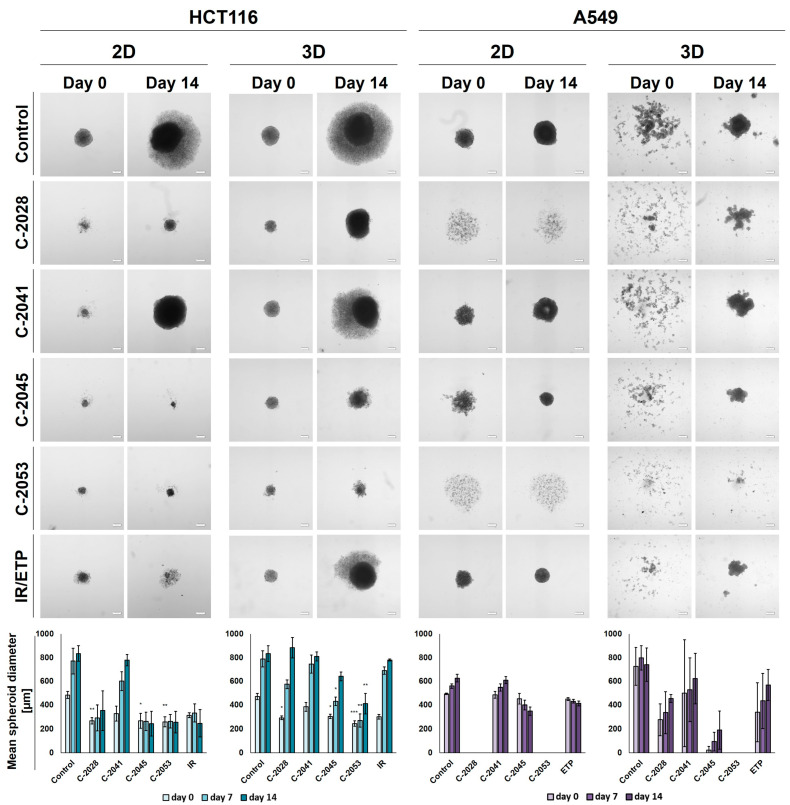
Spherogenic potential of the HCT116 (**left**) and A549 (**right**) cells cultured in 2D and 3D (secondary sphere formation) conditions after incubation with UAs and reference compounds. The cells cultured as monolayers and spheroids were incubated with C-2028, C-2041, C-2045, C-2053, and irinotecan (IR)/etoposide (ETP) at concentrations that corresponded to IC_90_ values (IC_50_ for IR/ETP) for 3 days, which were then were seeded onto ULA plates. Pictures of the generated spheroids were taken 3 days after seeding (day 0), and then 2 weeks later (day 14). Scale bar is 200 µm. The bar graphs show the quantified data, which are expressed as the mean diameters of the spheroids at day 0, day 7, and day 14. The data are presented as the means ± SDs of the 3 to 5 independent experiments. Statistical analysis was performed using Dunn’s nonparametric multiple comparisons test. Significant difference from control was set at * *p* < 0.05, ** *p* < 0.01, and *** *p* < 0.001. (*n* = 3–5).

**Figure 5 ijms-24-15780-f005:**
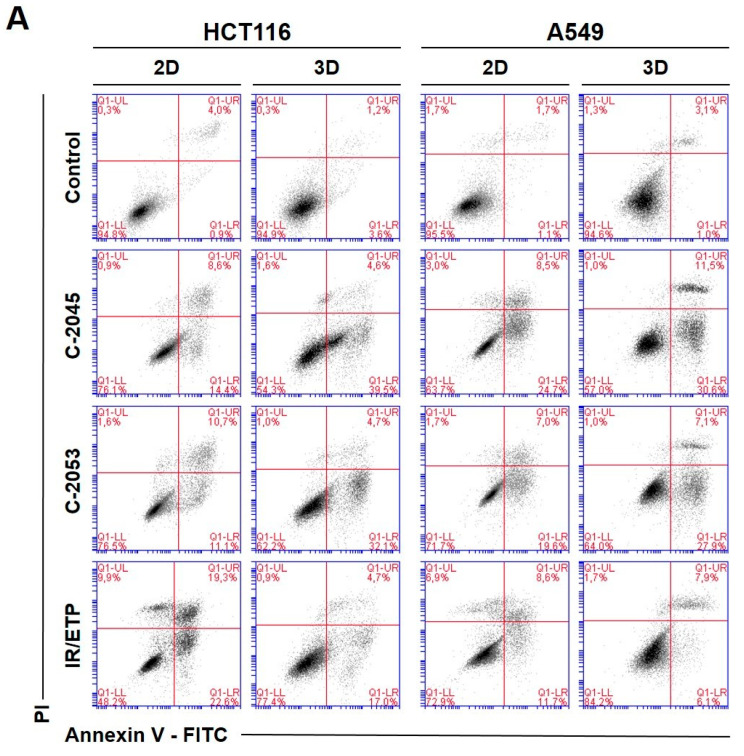
Phosphatidylserine externalization and membrane disruption in the HCT116 and A549 cells treated with C-2045, C-2053, and the reference compounds of irinotecan (IR) and etoposide (ETP). The cells were exposed to C-2045, C-2053, and IR/ETP at concentrations that corresponded to 5xIC_90_ values (5xIC_50_ for IR/ETP) for 72 h. These were then stained with annexin V–fluorescein isothiocyanate (FITC) and propidium iodide (PI), as well as analyzed using flow cytometry. (**A**) Representative bivariate flow cytometry histograms of the annexin V–FITC signal versus PI signal are shown. The bottom left quadrant represents the living cells (annexin V negative and PI negative); the bottom right quadrant represents the early-apoptotic cells (annexin V positive and PI negative); the top right quadrant represents the late-apoptotic cells (annexin V positive and PI positive); and the top left quadrant represents the primary necrotic cells (annexin V negative and PI positive). (**B**,**C**) These bar graphs show the quantified data, which are expressed as the percentages of HCT116 (**B**) and A549 (**C**) in the early-apoptotic, late-apoptotic, and necrotic cells in 2D and 3D cell cultures. The data are presented as the means ± SDs of the three to seven independent experiments. Statistical analysis was performed using Dunn’s nonparametric multiple comparisons test. Significant difference from the control was set at * *p* < 0.05, ** *p* < 0.01, and *** *p* < 0.001. (*n* = 3–7).

**Figure 6 ijms-24-15780-f006:**
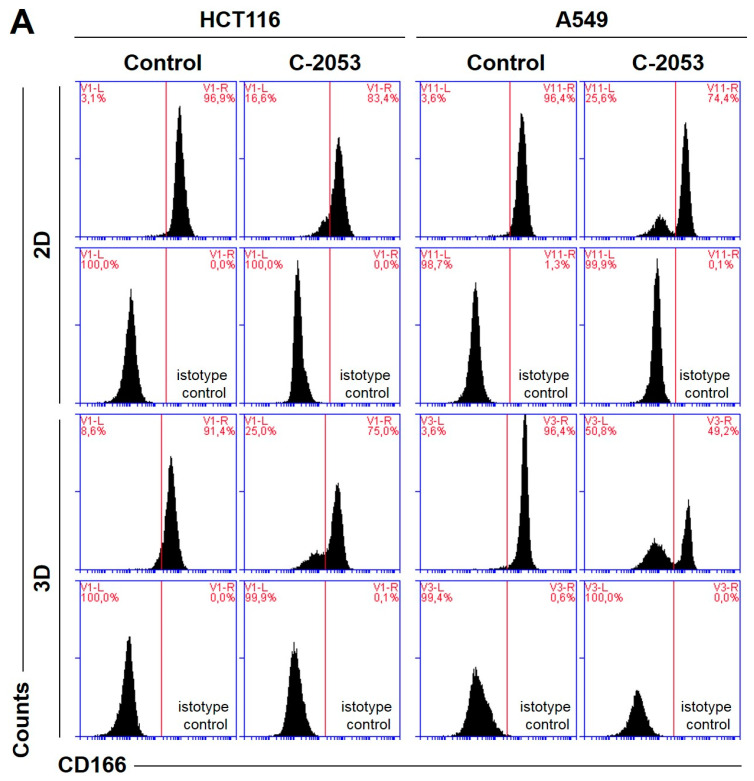
The levels of selected cancer stem cell markers in the HCT116 colon and A549 lung cancer cells that were cultured in monolayer conditions and as spheroids. The cells were treated with C-2045, C-2053, and the reference compounds (IR/ETP) at concentrations that corresponded to 5xIC_90_ doses (5xIC_50_ for IR/ETP). Then, the levels of CD166-, EpCAM-, CD44-, and CD133-positive cells were determined cytometrically. (**A**) Representative histograms of the level of CD166 in the HCT116 (**left**) and A549 (**right**) cells treated with C-2053 in 2D and 3D conditions (with isotype control). (**B**,**C**) Bar graphs presenting the quantified data, which are expressed as the percentages of HCT116 (**B**) and A549. (**C**) CD166^+^, EpCAM^+^, CD44^+^, and CD133^+^ cells in 2D and 3D cell cultures. The data are presented as the means ± SDs of the three to seven independent experiments. Statistical analysis was performed using Dunn’s nonparametric multiple comparisons test. Significant difference from the control was set at * *p* < 0.05, ** *p* < 0.01, and *** *p* < 0.001. (*n* = 3–7).

**Table 1 ijms-24-15780-t001:** Cytotoxicity of the UAs (C-2028, C-2041, C-2045, and C-2053) and the reference compound (etoposide) against the A549 cells. IC_90_ values for the UA and IC_50_ values for the reference compound. (*n* = 3–5).

	A549	
Compound	Drug Dose	Drug Concentration [µM]
C-2028	IC_90_	0.055 ± 0.010
C-2041	IC_90_	0.083 ± 0.021
C-2045	IC_90_	0.313 ± 0.044
C-2053	IC_90_	0.332 ± 0.070
Etoposide	IC_50_	5.587 ± 0.900

## Data Availability

Most of the data obtained during this study are included in this article (and the detailed Appendix A are available from the corresponding authors upon reasonable request).

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
