# Peer review of "Cellular Effects of Selected Unsymmetrical Bisacridines on the Multicellular Tumor Spheroids of HCT116 Colon and A549 Lung Cancer Cells in Comparison to Monolayer Cultures"

_ijms, 2023, doi:10.3390/ijms242115780_

Round 1

Reviewer 1 Report

Comments and Suggestions for Authors

This manuscript investigated the Cellular effects of selected unsymmetrical bisacridines on the 2 multicellular tumor spheroids of HCT116 colon and A549 lung 3 cancer cells in comparison to monolayer cultures.

Strengths

·       The manuscript is very well written and data are well interpreted. The subject area is hot and of interest.

·       Application of three-dimensional (3D) cultures of cancer cells, as in this study, more accurately mimic the 3D structure of malignant tissue and the microenvironment and are a better example of tumor cells in the body.

·       major experiments were conducted in order to reach reliable conclusion

·       Conclusion is consistent with the evidence.

Weakness

Please provide more justification for choosing Etoposide other than other anti-cancers for lung cancer cell line.

Author Response

This manuscript investigated the Cellular effects of selected unsymmetrical bisacridines on the 2 multicellular tumor spheroids of HCT116 colon and A549 lung 3 cancer cells in comparison to monolayer cultures.

Strengths

  • The manuscript is very well written and data are well interpreted. The subject area is hot and of interest.
  • Application of three-dimensional (3D) cultures of cancer cells, as in this study, more accurately mimic the 3D structure of malignant tissue and the microenvironment and are a better example of tumor cells in the body.
  • Major experiments were conducted in order to reach reliable conclusion
  • Conclusion is consistent with the evidence.

Weakness

Please provide more justification for choosing Etoposide other than other anti-cancers for lung cancer cell line.

We would like to thank the Reviewer for this comment. The selection of etoposide (ETP) as a reference compound in our studies is based on the fact, that this drug has been extensively studied for many years and therefore its mechanism of action is well known. Moreover, it is commonly applied in the treatment of lung cancer patients. Etoposide, a topoisomerase II inhibitor, is the most common therapeutic option for lung cancer, either alone or in combination with doxorubicin or cisplatin [P. Olszewska et al., Biomedicines 2022, 10(10), 2429].

Reviewer 2 Report

Comments and Suggestions for Authors

The authors present a comprehensive investigation of the effects of UAs on cancer stem cells. Their approach involves studying the effects of these compounds in both 2D and 3D cultures of HCT116 colon and A549 lung cancer cells. Initial results indicate that UAs show promise in targeting the viability of the cells studied, affecting their spherogenic potential, and potentially influencing the cancer stem cell-like population. The study is well designed, especially considering the potential therapeutic implications of UA.

Below are some specific comments and suggestions to further enhance the quality of the manuscript:

Comment: The introduction provides a broad overview of the importance of targeting CSCs in cancer therapy. The detailed discussion about the relevance of 3D cultures for CSC studies is commendable.

Suggestion: For a more comprehensive understanding, please include a brief overview of the current anticancer drugs that target CSCs. Emphasizing the gaps and the need for new drug candidates would enrich the context.

Citations in the Introduction:

Certain statements in the Introduction pertaining to the efficacy and toxicity of existing drugs that target CSCs could benefit from direct citations.

Suggestion: Kindly provide appropriate references to support these statements, offering readers a more in-depth insight and foundation for your study.

Visual Representations in Figure 3:

In Figure 3, representative images are missing. Given your results which show a significant effect, visual evidence would enhance clarity.

Suggestion: Please include pictures that display the number of colonies. This will greatly assist readers in visualizing and appreciating the impact of your findings.

I believe that addressing these suggestions will help in refining your manuscript for a broader audience. The work is commendable, and with a few improvements, it will be even more impactful.

Author Response

The authors present a comprehensive investigation of the effects of UAs on cancer stem cells. Their approach involves studying the effects of these compounds in both 2D and 3D cultures of HCT116 colon and A549 lung cancer cells. Initial results indicate that UAs show promise in targeting the viability of the cells studied, affecting their spherogenic potential, and potentially influencing the cancer stem cell-like population. The study is well designed, especially considering the potential therapeutic implications of UA.

Below are some specific comments and suggestions to further enhance the quality of the manuscript:

Comment: The introduction provides a broad overview of the importance of targeting CSCs in cancer therapy. The detailed discussion about the relevance of 3D cultures for CSC studies is commendable.

Suggestion: For a more comprehensive understanding, please include a brief overview of the current anticancer drugs that target CSCs. Emphasizing the gaps and the need for new drug candidates would enrich the context.

We kindly thank the Reviewer for this suggestion. A brief overview of current anticancer drugs targeting CSCs has been added to the manuscript.

Citations in the Introduction:

Certain statements in the Introduction pertaining to the efficacy and toxicity of existing drugs that target CSCs could benefit from direct citations.

Suggestion: Kindly provide appropriate references to support these statements, offering readers a more in-depth insight and foundation for your study.

We would like to thank the Reviewer for this feedback. As suggested, we provided references in this part.

Visual Representations in Figure 3:

In Figure 3, representative images are missing. Given your results which show a significant effect, visual evidence would enhance clarity.

Suggestion: Please include pictures that display the number of colonies. This will greatly assist readers in visualizing and appreciating the impact of your findings.

I believe that addressing these suggestions will help in refining your manuscript for a broader audience. The work is commendable, and with a few improvements, it will be even more impactful.

Representative images have been attached in supplementary materials. However, per Reviewer’s suggestion, we moved them to the manuscript, to better visualize our findings in the main text.